# Berberine, a Natural Compound That Demonstrates Antiviral Effects Against Ostreid Herpesvirus 1 Infection in *Anadara broughtonii*

**DOI:** 10.3390/v17020282

**Published:** 2025-02-18

**Authors:** Hui-Gang Kang, Mao-Le Wei, Jing-Li Wang, Cui-Ping Ma, Xiang Zhang, Bo-Wen Huang, Lu-Sheng Xin, Chang-Ming Bai

**Affiliations:** 1Qingdao Nucleic Acid Rapid Detection Engineering Research Center, Qingdao Key Laboratory of Nucleic Acid Rapid Detection, College of Biological Engineering, Qingdao University of Science and Technology, Qingdao 266042, China; kanghuigang0123@163.com (H.-G.K.); liuli000315@gmail.com (M.-L.W.); wjl19863763762@163.com (J.-L.W.); mcp169@163.com (C.-P.M.); 2State Key Laboratory of Mariculture Biobreeding and Sustainable Goods, Key Laboratory of Maricultural Organism Disease Control, Ministry of Agriculture, Yellow Sea Fisheries Research Institute, Chinese Academy of Fishery Sciences, Qingdao 266071, China; zhxiang1997@126.com (X.Z.); huangbw@ysfri.ac.cn (B.-W.H.); 3Laboratory for Marine Fisheries Science and Food Production Processes, Qingdao Marine Science and Technology Center, Qingdao 266237, China; 4Shandong Center of Technology Innovation for Oyster Seed Industry, Qingdao 266105, China

**Keywords:** OsHV-1, BBH, *Anadara broughtonii*, antiviral response

## Abstract

Ostreid herpesvirus 1 (OsHV-1) infection is the primary viral disease responsible for large-scale mortality in bivalve mollusks worldwide, and effective strategies to control the outbreaks of this disease are still lacking. Berberine (BBR), a plant-derived alkaloid, has demonstrated antiviral activity against various vertebrate viruses, while its potential antiviral effects on molluscan herpesviruses remain to be fully elucidated. Therefore, the present study sought to investigate the potential of berberine hydrochloride (BBH) against OsHV-1 infection in blood clams (*Anadara broughtonii*). The most optimal BBH concentration was figured out according to virus replication and mortality rates during in vivo experimental infection. Quantitative PCR and reverse transcription quantitative PCR were utilized to monitor the OsHV-1 genomic copy numbers and viral gene transcription levels during the development of OsHV-1 infection in the BBH-treated and control groups. The results demonstrated that a 3 mg/L BBH bath immersion significantly suppressed OsHV-1 replication in blood clams. During the early stage of infection (24 h), BBH treatment significantly reduced the expression of OsHV-1 open reading frames (ORFs) related to early enzymes, putative membrane proteins, and nucleocapsid proteins. At 96 h post-infection, all untreated blood clams died, whereas the survival rate of BBH-treated individuals increased to 46.67%. This study provides preliminary evidence for the inhibitory effects of BBH on OsHV-1, paving the way for the development of pharmacological control technologies for OsHV-1 infections.

## 1. Introduction

Ostreid herpesvirus 1 (OsHV-1), a member of the family *Malacoherpesviridae* and genus *Ostreavirus*, is one of the most prevalent viral pathogens affecting bivalves on a global scale [1]. Since its discovery, OsHV-1 has been responsible for significant economic losses in the shellfish aquaculture industry [2,3]. Bivalves represent a pivotal group of marine organisms, playing a crucial role in marine ecosystems and supporting important fisheries and aquaculture industries worldwide [4,5]. The first documented cases of herpesvirus infections in invertebrates occurred in adult oysters along the U.S. East Coast in 1972 [6]. In 1999, herpesvirus particles linked to high mortality were first successfully purified from juvenile oysters in France [7]. The presence of OsHV-1 has been detected in various bivalve species (e.g., *Crassostrea gigas*, *Ostrea angasi*, *Ruditapes decussatus*, *Pecten maximus*) across Europe, North America, Australia, and other regions, leading to large-scale outbreaks and mortality events [8,9,10,11]. Several variants of OsHV-1, including OsHV-1μvar, Acute Viral Necrotic Virus (AVNV) of scallops, and OsHV-1-SB, have emerged globally, posing significant challenges to the control of molluscan herpesvirus infections [12,13,14,15,16,17]. Research indicates that OsHV-1 infection compromises the antioxidant system in oysters, causing cellular damage, exacerbating immunosuppression, and ultimately triggering opportunistic bacterial infections [18,19]. OsHV-1 outbreaks typically occur during the warmer summer months, with mortality rates in infected juvenile Pacific oysters often exceeding 80% [20,21]. In recent years, *Anadara broughtonii* has been identified as a susceptible host of OsHV-1, with mass mortality events linked to this viral infection [15]. Since the initial detection of OsHV-1 in *A. broughtonii* in 2012, it has become the foremost cause of large-scale mortality in this species [22]. Despite the advancement in knowledge of OsHV-1, there is currently no efficient or cost-effective treatment to combat this viral disease. Consequently, there is an urgent need to explore and develop effective control interventions to mitigate the economic losses incurred as a result of this viral disease.

Berberine (BBR), an isoquinoline alkaloid, is a natural compound that occurs in various medicinal plants, including *Coptis* chinensis, *Phellodendron amurense*, and *Berberis* species [23]. BBR has a long history in traditional herbal medicine, where it is frequently used to treat gastrointestinal inflammation in humans [24]. BBR has low toxicity and minimal side effects, demonstrating unique biochemical and pharmacological activities, including antibacterial, antiviral, anti-inflammatory, anticancer, and metabolic regulatory effects [25]. The molecular structure of BBR consists of a tetracyclic system with an N-benzyl tetrahydroisoquinoline backbone and an additional C13 carbon bridge. The complexity of this structure allows BBR to interact with multiple biological targets, thereby exerting its pharmacological effects [26]. Berberine hydrochloride (BBH) is the hydrochloride salt form of BBR. This form of BBR is able to effectively penetrate the cell membranes and enter cells, which is a factor in its wide-ranging utilization due to its high bioavailability. BBR has been shown to possess significant antiviral activity against a range of viruses, including herpes simplex virus (HSV), influenza virus, human cytomegalovirus (HCMV), and enterovirus [27,28,29]. The primary mechanism of action of BBR involves its interaction with intracellular targets, particularly enzymes and DNA. Studies have shown that BBR inhibits HCMV replication by interfering with the transactivation activity of the immediate-early 2 (IE2) protein [27]. Furthermore, BBR has been demonstrated to act early in the replication cycle of HSV, inhibiting HSV-induced NF-κB activation and c-Jun N-terminal kinase (JNK) phosphorylation [28]. Additionally, BBH has been reported in studies on viral diseases in aquatic animals. For example, BBH has been shown to inhibit viral gene transcription and to suppress Cyprinid herpesvirus 2 (CyHV-2) replication in RyuF-2 cells [30]. However, the effects of BBH on molluscan herpesviruses remain largely unexplored.

The present study evaluated the efficacy of BBH in suppressing OsHV-1 replication through bath treatments administered to OsHV-1-infected blood clams. The findings of this study seek to assess the potential of BBH as an antiviral agent for the prevention and control of mollusk diseases in aquaculture.

## 2. Materials and Methods

### 2.1. Drugs and Animals

The berberine hydrochloride (BBH) capsules were obtained from Yunnan Baiyao Group Co., Ltd., Kunming, China, with a dosage of 0.1 g per capsule. The experimental animals, *Anadara broughtonii*, were a batch of 300 healthy blood clams (shell length 54.86–67.93 mm) purchased in January 2024 from a bivalve farm in Rizhao, Shandong Province, China. These blood clams were caught from the wild population in the Yellow Sea. Following their transportation to the aquaculture facility of the Yellow Sea Fisheries Research Institute (YSFRI), the blood clams were accommodated in 100 L tanks (10 blood clams per tank), with aerated, filtered seawater supplied and half of the water replaced daily. The animals were provided with sufficient bivalve feed. Throughout the experiment, the salinity and temperature of the seawater were maintained at 30 ± 1 ppt and 18 ± 1 °C, respectively. Following a 2-week acclimation period, 30 blood clams were randomly selected, all of which tested negative for OsHV-1. Prior to the commencement of the experiment, the blood clams were subjected to 2-week fasting, a condition that was intended to induce a state of starvation followed by maximum filtration activity [31]. The experiment was conducted in strict accordance with the ethical guidelines of the Animal Ethics Committee of the Yellow Sea Fisheries Research Institute, CAFS.

### 2.2. OsHV-1 Infection Experiment

The tissue homogenate was prepared using the standard inoculation protocol described by Schikorski et al. [32]. Briefly, the mantle tissues of OsHV-1-infected blood clams were collected and homogenized in sterile seawater. The mixture was centrifuged briefly at 1000 rpm and 4 °C, and the supernatant was retained. The resulting viral inoculum was filtered sequentially through filters with pore sizes of 5 μm, 2 μm, 0.45 μm, and 0.22 μm, then stored at 4 °C for later use. Due to the small and concealed adductor muscle of the blood clams, the viral inoculum was injected into the foot, which is exposed and easier to handle. In the infection experiment, the blood clams were anesthetized with 70 g/L MgCl_2_, and 100 μL of the viral inoculum (adjusted to 1 × 10^4^ copies/μL) was injected into the foot of each blood clam.

### 2.3. BBH Medicated Bath

Since blood clams are filter-feeding animals, we adopted an immersion method for drug administration. To ensure the complete dissolution of the drug in seawater, berberine hydrochloride was sonicated. The employment of ultrasonic technology has been demonstrated to enhance the solubility of drugs, particularly those that are inherently water-insoluble. This method accelerates dissolution and enhances drug bioavailability [33]. Berberine hydrochloride powder was added to a 50 mL centrifuge tube with 30 mL of sterile seawater and placed in a sonicator (model: SB25-12D, manufacturer: NingBo Scientz Biotechnology Co., Ltd., Ningbo, China) for 10 min at 20 kHz and subjected to intermittent shaking (at a frequency of 5 min) during the sonication process to ensure complete dissolution of all particles, and this was continued until no visible undissolved particles were observed. In order to enhance the absorption of the drug by the blood clams, an innovation was introduced in the method of drug delivery. Concentrated diatoms, used to feed the blood clams, were added to the sonicated BBH solution, facilitating complete integration of the drug and promoting its adsorption onto the diatoms [34].

To determine the safe drug concentration of BBR in the blood clams [35], different amounts of berberine hydrochloride (0.1 g, 0.3 g, 0.6 g, and 0.9 g) were added to the tanks, resulting in final concentrations of 1 mg/L, 3 mg/L, 6 mg/L, and 9 mg/L, respectively, according to previous reports and the weight of the blood clams [30]. After culturing for 7 days under identical conditions, we performed a preliminary toxicity assessment at the individual level by observing the behavior, appearance, and mortality of blood clams treated with different drug concentrations. Both the protective and therapeutic effects of BBH on vertebrate herpesviruses have been reported previously [27,28,29,30]. To determine the most effective measure for OsHV-1 control, BBH bath was administered both before and after experimental infection. The blood clams were randomly assigned to enter the medicated bath group or the control group. One day prior to the injection of OsHV-1, the prepared berberine hydrochloride was distributed evenly in the tanks as a bath for the blood clams. After the viral infection, berberine hydrochloride was administered continuously to the group tanks containing the medicated bath, with the seawater being replaced every 24 h and the drug being re-administered. No drug exposure was applied to the control group.

### 2.4. Tissue Sample Collection and Mortality Assessment

Five tanks with 10 clams in each tank were assigned to each treatment group, with three tanks designated for the collection of tissue samples and the remaining two for the monitoring of mortality. Three blood clams from each medicated bath group and the control group were collected at 12, 24, 36, 48, 60, and 72 h. The blood clams were dissected, and hemolymph was collected from the adductor muscle sinus of each blood clam using a 23 G needle and a 5 mL syringe [36]. The hemolymph from each group was divided into three equal portions, after which it was subjected to centrifugation at 800 rpm for 5 min at 4 °C. The supernatant was discarded. Then, 1 mL of Trizol (Invitrogen, Carlsbad, CA, USA) was added to two aliquots of each sample which were then immediately frozen in liquid nitrogen; all samples were stored at −80 °C for RNA and DNA extraction and further analysis [37].

### 2.5. DNA Extraction and Quantification of OsHV-1 DNA

DNA was extracted from frozen hemocyte samples using the TIANamp™ Marine Animal DNA Kit (Tiangen Biotech, Beijing, China) according to the manufacturer’s protocol. DNA purity and quantity were assessed using a Nanodrop 2000 spectrophotometer (Thermo Scientific, Waltham, MA, USA). The OsHV-1 DNA copy number was determined using quantitative PCR (qPCR) based on the protocol published by Bai et al., as fully described in the references [15]. Briefly, the amplification was performed in a 20 µL reaction mixture, including 10 µL of 2× FastStart Essential DNA Probe Master Mix (Roche), 0.8 µL of each primer (10 µM), 0.4 µL of TaqMan^®^ Probe (10 µM), 1.6 µL of template DNA, and 6.4 µL of DEPC water. Each DNA sample was quantified using the CFX Connect™ Real-Time System (Bio-Rad Laboratories, Inc., Hercules, CA, USA), with each qPCR analysis performed in triplicate. The OsHV-1 DNA copy number was evaluated as the average number of OsHV-1 DNA copies per ng of total DNA.

### 2.6. Total RNA Extraction and cDNA Synthesis

Total RNA was extracted from 50 mg of each tissue sample using TRIzol reagent (Invitrogen) according to the manufacturer’s protocol. cDNA was synthesized using gDNA Eraser-treated total RNA as a template, according to the guidelines for reverse transcription PCR (RT-PCR) and reverse transcription reagents (TaKaRa, Shiga, Japan). The synthesis reaction was performed at 37 °C for 15 min, followed by heating at 85 °C for 5 s to terminate the reaction. The cDNA mixture was diluted at a ratio of 1:10 and stored at −80 °C.

### 2.7. RT-qPCR Measurement of Relative Expression Levels of OsHV-1 ORFs

Real-time quantitative PCR (RT-qPCR) analysis was conducted using the Bio-Rad CFX Connect Real-Time System (Bio-Rad Laboratories), with each RT-qPCR sample analyzed in triplicate. The total amplification reaction volume was 20 μL, consisting of 5 μL of diluted cDNA template (1:10), 10 μL of 2× SYBR Green PCR Mix (Takara), 0.8 μL of forward and reverse primers (250 nM), and 3.4 μL of distilled water. The cycling conditions were as follows: 95 °C for 1 min, followed by 95 °C for 10 s, and 60 °C for 30 s for a total of 40 cycles. The specificity of the primers was assessed using a melt curve, where the temperature was increased from 60 °C to 95 °C (at 0.5 °C per 10 s). The forward and reverse primers were designed using Primer Premier 5 software (PREMIER Biosoft) and synthesized by Genechem Co., Ltd. (Shanghai, China).

Based on the expression analysis of blood clam housekeeping genes under OsHV-1 challenge, EF-1α and RL15 were selected as housekeeping genes for the RT-qPCR normalization of viral gene expression. The Ct difference between viral genes and EF-1α provides the relative expression levels of the viral genes [38]. To investigate viral gene expression, 26 genes were selected from the 124 ORFs of OsHV-1 based on their protein function or structure. These genes fall into five gene/family categories (Table 1) [1,39]: (i) seven ORFs encoding enzymes (ORF 7, ORF 20, ORF 27, ORF 34, ORF 49, ORF 51, ORF 67); (ii) four ORFs of the RING gene family (ORF 9, ORF 42, ORF 53, ORF 87, with ORF 42 and ORF 87 containing a BIR domain) [26]; (iii) twelve ORFs encoding membrane proteins (ORF 16, ORF 25, ORF 36, ORF 41, ORF 54, ORF 57, ORF 59, ORF 68, ORF 72, ORF 80, ORF 84, ORF 88); (iv) one ORF encoding a capsid protein (ORF 104); and (v) two ORFs encoding unknown proteins (ORF 11, ORF 37). Table 1 provides primer information for the selected OsHV-1 ORFs and blood clam housekeeping genes used in this RT-qPCR experiment.

### 2.8. Data Analysis

Survival rates, viral genome copy numbers, and relative expression levels of OsHV-1 ORFs were compared between the medicated bath group and the control group (non-drug exposed) at various time points of OsHV-1 infection (12 h, 24 h, 36 h, 48 h, 60 h, and 72 h). The 2^−∆∆CT^ method was applied to accurately analyze the relative expression levels of ORFs [40]. Blood clam housekeeping genes (Table 1) were used as internal references to normalize the relative expression levels between samples. The average relative expression for each group was calculated using the 2^−∆∆CT^ method, with the control group used for normalization. The relative expression results of ORFs are expressed as the mean ± standard error. Statistical analysis was performed using IBM SPSS Statistics 19.0, and the relevant data were plotted using GraphPad Prism software (version 10.2.1).

## 3. Results

### 3.1. Survival Rate of Blood Clams

Following the injection of the viral inoculum into the blood clams, mortality rates were monitored at 12 h intervals in order to assess the survival of blood clams in the BBH bath groups (with concentrations of 1 mg/L, 3 mg/L, 6 mg/L, and 9 mg/L) and the control group (without drug exposure). Following OsHV-1 infection, the survival rates of blood clams in the BBH bath groups were higher than those in the control group. Among them, blood clams infected with OsHV-1 and treated with a 3 mg/L BBH bath exhibited the most significant therapeutic effect, with a survival rate of 46.67% at 96 h post-infection (Figure 1). However, the control group, which received no drug treatment, showed the first signs of mortality at 36 hpi. By 60 hpi, the remaining blood clams were in a moribund state, and the survival rate had dropped to zero by 72 h. In contrast, the 3 mg/L BBH group showed mortality only after 60 hpi.

### 3.2. OsHV-1 DNA Copy Number

Virus DNA quantification was performed on the 3 mg/L BBH bath group and the control group at different time points post-OsHV-1 infection. The qPCR results showed significant differences (*p* < 0.05) in OsHV-1 DNA copy numbers between the medicated bath group and the control group at 12 h, 24 h, 36 h, 48 h, 60 h, and 72 h post-infection. The bath with a concentration of 3 mg/L BBH led to a substantial reduction in OsHV-1 replication (Figure 2). In the control group, a significant increase in OsHV-1 viral particle growth was observed, primarily during the mid-infection period (48 h). In contrast, the medicated bath group exhibited a delay in viral particle replication, with a significant increase in viral particle production only beginning at the later stages of infection (72 h). These findings suggest that the BBH bath can effectively interfere with OsHV-1 viral DNA replication in infected blood clams, particularly during the early stages, where viral particle proliferation is significantly inhibited. Symptoms generally appeared when the viral load reached a threshold (generally considered to be around 10^4^ copies/ng DNA) [41]. In the 3 mg/L BBH treatment group, the viral load remained below this threshold until 60 hpi, which explains why mortality did not occur in this group until after 60 hpi.

### 3.3. OsHV-1 ORFs Relative Expression

In this study, a selection of 26 OsHV-1 ORFs from five genomic/family groups was made on the basis of their protein function, structure, and gene family classification (Table 1). The relative expression of these ORFs was then assessed in the 3 mg/L BBH bath group and the control group using real-time RT-PCR. The results showed that at 24 h post-infection (early stage), the expression levels of OsHV-1 ORFs were significantly reduced in the BBH bath group compared to the control group (*p* < 0.05). This finding is consistent with the results of the OsHV-1 DNA copy number assay. The variation in the relative expression of OsHV-1 ORFs between the BBH bath group and the control group was more evident at 24 h post-infection (early stage) than at 72 h (late stage) (Figure 3).

It is noteworthy that specific ORFs encoding enzymes or proteins fulfill pivotal functions in the process of viral particle replication. For example, ORF7 is hypothesized to encode a member of the DNA primase family, ORF25, ORF36, ORF72, and ORF80 are hypothesized to encode membrane proteins, and ORF104 is hypothesized to encode the nucleocapsid protein. A statistically significant disparity in the relative expression levels of these ORFs (*p* < 0.05) was observed between the BBH bath group and the control group (Figure 4). Furthermore, the expression levels of different ORFs at the same time point (24 h post-infection) also showed distinct variations, which may be related to the characteristics of the proteins they encode [42].

## 4. Discussion

A substantial body of research has been dedicated to the exploration of the antiviral properties of numerous herbal medicines in Traditional Chinese Medicine (TCM). The bioactive molecules present in these herbs, including plant alkaloids, flavonoids, and terpenoids, are recognized as pivotal elements contributing to their therapeutic effects [43]. Plant-derived alkaloids have been shown to be effective in reducing viral load by interfering with viral adsorption and invasion of host cells, as well as by modulating immune responses to enhance the host’s antiviral capacity. Berberine (BBR), a natural alkaloid with broad-spectrum antiviral activity, has demonstrated significant potential in the control of viral diseases in aquatic animals. Berberine has typically been used by oral administration in fish and has shown a dose-dependent antiviral effect [30]. Due to the unique feeding habits of bivalves, we administered BBH by bath treatment and also demonstrated the inhibitory effect on OsHV-1 replication, although the benefit of the drug is not dose dependent in the present study. We proposed that the solubility of BBH in seawater was responsible for the unexpected results. Fine particles could be detected in tanks with concentrations of 6 mg/mL and 9 mg/mL BBH. These particles may interfere with the filter feeding of clams, and thus the treatment effect.

The mechanism by which BBR exerts its antiviral effects is through the modulation of inflammation, oxidative stress, and immune responses; one such key mechanism is the regulation of the NF-κB pathway. BBR has been shown to downregulate virus-induced NF-κB activation and prevent the degradation of the endogenous NF-κB inhibitor IκBα [44]. BBH may help to protect crucian carp from CyHV-2 infection by negatively regulating the NF-κB signaling pathway and its downstream inflammatory cytokines [45]. BBR has demonstrated therapeutic potential in the treatment of Severe Acute Respiratory Syndrome Coronavirus (SARS-CoV). During SARS-CoV infection, the NF-κB signaling pathway is activated by viral proteins such as nsp1, nsp2, nsp7, and the spike protein, and BBR has been shown to inhibit this pathway [46]. In addition to this, BBR has been shown to exert anti-neuroinflammatory effects and neuroprotective properties in the central nervous system (CNS). It has been demonstrated that BBR can inhibit the excessive production of pro-inflammatory cytokines and nitric oxide (NO), thereby protecting the CNS from neurodegenerative diseases by suppressing neuroinflammation and improving neurological outcomes [23]. BBR has antiviral preventive effects, which are crucial for pre-treatment and for enhancing therapeutic efficacy. Studies show that BBR regulates ROS production and inhibits viral replication by downregulating autophagy, thereby preventing fatal enterovirus 71 (EV71) neuroinfection [47]. In addition, Hong et al. demonstrated that BBR inhibits Hepatitis C virus (HCV) replication by targeting the binding of the E2 glycoprotein and blocking HCV attachment and entry, thus suggesting that BBR may be a promising candidate for developing entry inhibitors to prevent and treat HCV infection [48]. Furthermore, BBH has been shown to be capable of systemically inhibiting CyHV-2 gene transcription and suppressing its replication in RyuF-2 cells, thereby demonstrating a dose-dependent protective effect against CyHV-2 infection in crucian carp [30]. Specifically, we introduced a BBH bath treatment method with the aim of effectively inhibiting OsHV-1 replication in blood clams.

Proteins encoded by specific ORFs in the OsHV-1 genome play a crucial role in viral invasion and early replication. They facilitate the specific binding of the virus to host cells and promote viral replication within host cells, leading to viral proliferation. For example, the putative membrane proteins encoded by ORF25, ORF41, and ORF72 appear to be involved in the interaction between OsHV-1 and host cells, and these membrane proteins are believed to be related to the attachment of the virus to oyster cells, facilitating viral entry into host cells [49]. Furthermore, the analysis of the interactions between viral membrane proteins and host–cell surface molecules has shown that ORF25 and ORF72 can interact synergistically with actin and tubulin, respectively [50]. Morga et al. were the first to report the detection of viral DNA in Pacific oyster hemocytes 1 h post-exposure, with significant increases at 18 and 24 h. Viral transcript levels also varied depending on the analyzed ORFs [42]. Furthermore, it has been hypothesized that certain OsHV-1-encoded proteins may also possess immunomodulatory functions, thereby enabling the virus to evade the host immune system by modulating immune responses. Two dUTPase-like proteins (ORF 27 and ORF 75) of OsHV-1 are highly expressed at the transcriptomic level, with the potential to disrupt the immune response of Pacific oysters and inhibit host antiviral responses, thus enhancing viral survival [51]. Furthermore, the apoptosis inhibitory protein (IAP) family members ORF87 and ORF106 have been shown to maximize the amplification of viral particles in the early stages of infection by inhibiting TNF-mediated cell apoptosis in Pacific oyster cells [52]. Previous studies have reported that OsHV-1, ORF42, and ORF87 are significantly upregulated in the early stages of infection (4 h post-infection), limiting the host–cell apoptotic response [39]. Recent studies using long-read sequencing and multi-omics analysis techniques have facilitated the in-depth analysis of key protein-coding genes in the OsHV-1 genome, with a particular focus on the conservation of transcriptional structures related to the major capsid protein (MCP, ORF104) and capsid maturation protease (ORF107) isoforms, further revealing the conserved expression strategies of the viral capsid maturation module and the virus’s evasion of the host ADAR antiviral defense mechanism [53]. In this study, particularly during the initial phases of infection (24 h), the relative expression of OsHV-1 ORFs was considerably diminished in blood clams subjected to BBH bath treatment. This observation indicates that BBH may potently impede OsHV-1 invasion and early replication in blood clams, thereby decelerating the progression of the disease. While BBH has been shown to be effective in increasing the resistance of blood clams to OsHV-1 infection, further research is needed to understand the molecular mechanisms by which BBH regulates the resistance of blood clams to OsHV-1.

## 5. Conclusions

The present study demonstrates the efficacy of BBH in suppressing OsHV-1 replication in *Anadara broughtonii*, particularly during the initial phases of infection. This inhibition is achieved by blocking the transcription of early viral genes and membrane protein genes involved in viral attachment, thereby markedly enhancing the antiviral response of *A. broughtonii* and increasing its survival rate. This finding provides scientific evidence to support the potential application of BBH as an antiviral agent in the aquaculture industry. In the future, we must further validate BBH’s antiviral effect against OsHV-1 infection in mollusk production.

## Figures and Tables

**Figure 1 viruses-17-00282-f001:**
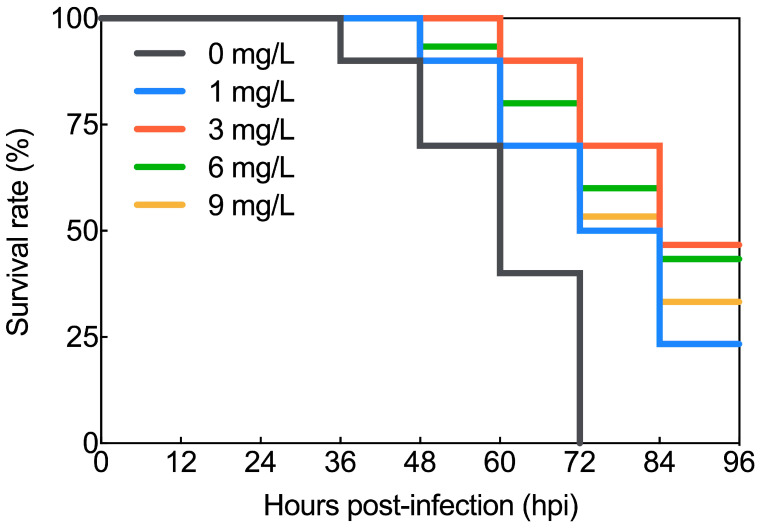
Survival rate of OsHV-1-infected blood clams treated with various concentrations of BBH. At 96 h post-OsHV-1 infection, the survival rates of blood clams treated with 1 mg/L, 3 mg/L, 6 mg/L, and 9 mg/L BBH baths were 23.33%, 46.67%, 43.33%, and 33.33%, respectively, while the survival rate of blood clams in the control group (without drug exposure) was 0%.

**Figure 2 viruses-17-00282-f002:**
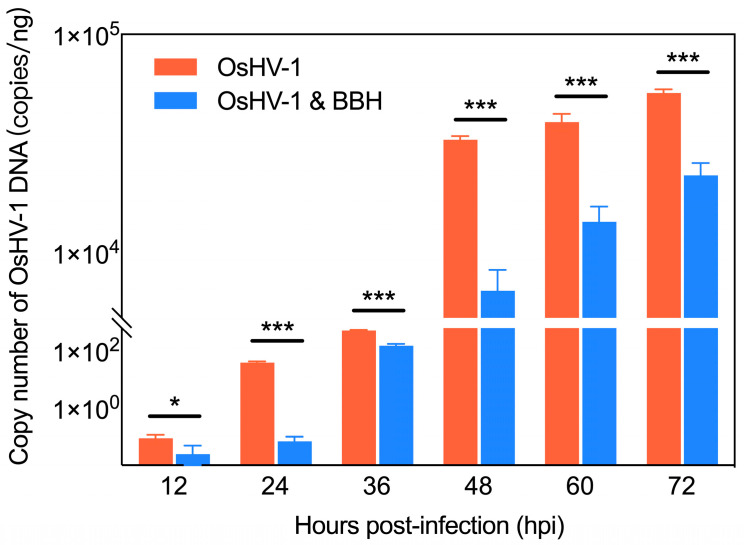
OsHV-1 DNA copy numbers in the 3 mg/L BBH bath group and the control group. The average OsHV-1 DNA copy numbers (copies/ng DNA) in the control group and the medicated bath group from 12 to 72 h post-infection were as follows: 0.08 vs. 0.02 (12 h), 37.19 vs. 0.07 (24 h), 4.97 × 10^2^ vs. 1.43 × 10^2^ (36 h), 3.28 × 10^4^ vs. 6.67 × 10^3^ (48 h), 3.97 × 10^4^ vs. 1.38 × 10^4^ (60 h), and 5.38 × 10^4^ vs. 2.25 × 10^4^ (72 h). * *p* < 0.05 and *** *p* < 0.001.

**Figure 3 viruses-17-00282-f003:**
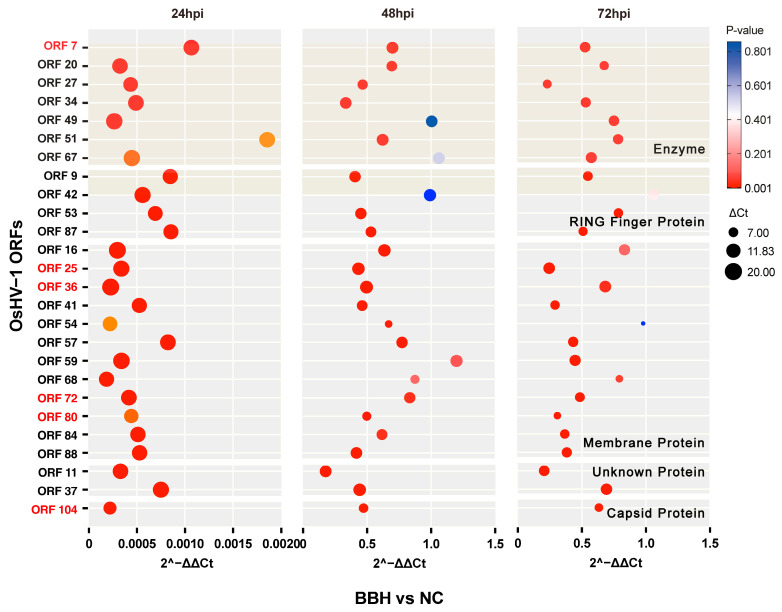
Bubble chart showing the relative expression levels of 26 OsHV-1 ORFs in the 3 mg/L berberine hydrochloride immersion group and the control group. At 24, 48, and 72 h post-infection, the relative expression levels of 26 OsHV-1 ORFs from five genomic/family groups were measured in both the 3 mg/L berberine hydrochloride immersion group and the control group. The color indicates the significance level of gene expression, with red representing higher significance and blue representing lower significance.

**Figure 4 viruses-17-00282-f004:**
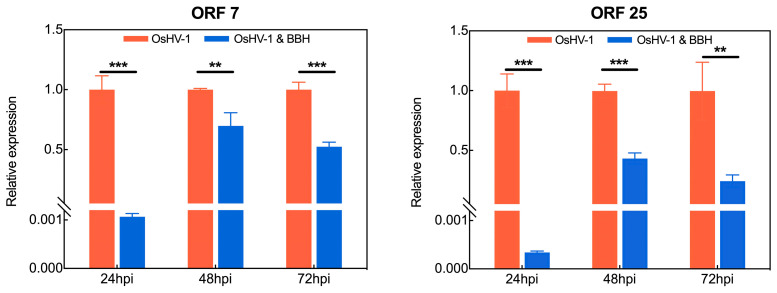
Bar graph showing the relative expression levels of OsHV-1 ORFs. At different time points of viral infection (24 h, 48 h, and 72 h), the relative expression levels of OsHV-1 ORF7, ORF25, ORF36, ORF72, ORF80, and ORF104 in the experimental group (3 mg/L berberine hydrochloride treatment) and the control group (no drug exposure) were normalized for comparison. Statistical significance was assessed using multiple t-tests, with * *p* < 0.05, ** *p* < 0.01, and *** *p* < 0.001.

**Table 1 viruses-17-00282-t001:** Primer information for OsHV-1 open reading frames (ORFs) and blood clam housekeeping genes used in RT-qPCR.

Gene Name	Forward Primer (5′–3′)	Reverse Primer (5′–3′)	Amplicon Length (bp)	Molecular Function
ORF7	CGATAGCGAGGGAGACGATG	CCGTGTGTTCCATTCTGCAA	198	DNA primase (Enzyme)
ORF9	CCATCCTCGTTGGGGGTAAT	CGTGGCATGTGAGCAGAGAG	189	RING finger protein
ORF11	GGCGTGGAGAGTTCATCAAGGT	TTGCCGCATCTGCCAATGGT	201	Unknown protein
ORF16	TCCAAACCCAGACCCAGCTT	CCAAAGAGCAGGCCAGGGTA	148	Membrane protein
ORF20	CTGGCTGTTCTGCCATTTCC	AGACAGCGGCAAGGTGATGT	213	Ribonucleotide reductase (Enzyme)
ORF25	CTCGCCAAAGGTCGTATCCA	CCACAAGGGTGAATTCCATGTT	200	Class I membrane protein
ORF27	ACCGCCCATGGTGCTAATTC	CCTGCAAGGCCAAGTATGGA	189	dUTPase (Enzyme)
ORF34	CTTGGACAAATGCATCCCAAA	TGGCACCCACATTCAAGGTT	183	dUTPase (Enzyme)
ORF36	AGAGACAGGCATGGCAGAGAGT	ACTTCTTGGAATCGGTCCTCGG	122	Membrane protein
ORF37	CAACCGACAATTCCCTCCCGAT	CACCAGCCCGCGTTAAGACAT	122	Unknown protein
ORF41	AACCGTGTGTCCCACCATTC	ACCGCAGTGCAACTCCAATC	183	Class I membrane protein
ORF42	GCAGGCATAACAGGTGAGCA	TGAGAGGCGTGACAGGGAAT	205	BIR protein containing RING finger
ORF49	ACCCAAAGTCGGTGGGGTAA	CGGAGATTTGCGGTGTACCA	191	Helicase (Enzyme)
ORF51	CGGCAAGGGAGATTGACCACAA	ACCAATTCAGGCACTGGCTTCC	268	Class I ribonucleotide reductase (Enzyme)
ORF53	CCGAAAAACCAGGGACTGGA	TGGGCGGGAAGTAGATCGTT	197	RING finger protein
ORF54	TAACCATGGGCACCGAAGAC	ACATCTCGCCGCTTTGTCAC	193	Class I membrane glycoprotein
ORF57	TTACCAGCACCGAGCAGGAT	TCGCCGCTTTTATCCAACAC	150	Multiple transmembrane protein
ORF59	ACGATGACGGAGATGGAGTTGT	AGTGATTGGAGGTTGTGGAGGT	281	Multiple transmembrane protein
ORF67	GATGACGGCATTGGTGAGGT	CCTGTGTTGCGGCTTGGTTA	195	SF2 helicase (Enzyme)
ORF68	TTGGTACCGGCCATTGACAT	GGAACCGAGTTTGCCACCTT	185	Class I membrane glycoprotein
ORF72	ACCTCCCCGTCAATGGTATGA	TCCACCACACCCCTACAATCA	180	Membrane protein
ORF80	GGATTGGTGCAACAGTATTGGC	TTCGATGCCTTGCCGGTGAT	130	Membrane protein
ORF84	CATCTGTGCTGTCATCGTCCA	TGTACACCTCGGTGCAGGTTTT	186	Membrane protein
ORF87	CACAGACGACATTTCCCCAAA	AAAGCTCGTTCCCACATTGGT	196	BIR protein containing RING finger
ORF88	CCAACAGATGTGGTGAGCGGTA	TGGTGGCGGTGGTGGTGAAT	152	Class I membrane protein
ORF104	AGGCAGCCATTGAAGCAGAGTC	GCGGTGTTGTCAGAGGTGGTAA	289	Capsid protein
EF-1α	TTAGACGCAATCCTCCCTCC	TAACGACGGTTCCTGGTTTG	142	Housekeeping gene
RL15	AGACCAGACAAAGCCAGAAGAC	GCTGAAGTAAGTCCACGCATT	371	Housekeeping gene

## Data Availability

The authors confirm that the data supporting the findings of this study are available within the article.

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
