# Peer review of "Berberine, a Natural Compound That Demonstrates Antiviral Effects Against Ostreid Herpesvirus 1 Infection in Anadara broughtonii"

_viruses, 2025, doi:10.3390/v17020282_

Round 1
Reviewer 1 Report
Comments and Suggestions for Authors
Major comments:
1. This study identified the suppression of OsHV-1 by BBH, but the mechanistic details were not explored. It is therefore remains unclear whether the antiviral effects of BBH are direct or indirect.
2. The study uses multiple concentrations of BBH but it is unclear why higher concentrations (6 and 9 mg/L) did not improve survival significantly. The preliminary toxicity data of BBH is missing.
3. Only some of the 124 OsHV-1 ORFs were analyzed, it may not capture all relevant viral dynamics.
4. The heatmap figure is not clearly labeled.
Minor comments:
1. What are the study's limitations? The lack of field validation or molecular mechanistic studies were not discussed in the discussion section.
2. How sample sizes were determined is not explained.
3. There are two Figures labeled as Figure 2.
Comments on the Quality of English LanguageThe manuscript would benefit from professional language editing to enhance clarity, readability, and overall quality.
Reviewer 2 Report
Comments and Suggestions for Authors
Reviewer’s comments on the manuscript:
Berberine, a natural compound with antiviral effects against 2
Ostreid herpesvirus 1 infection in Anadara broughtonii
Hui-Gang Kang, Mao-Le Wei, Jing-Li Wang, Cui-Ping Ma, Xiang Zhang, Bo-Wen Huang, Lu-Sheng Xin, Chang-Ming Bai
This manuscript presents novel and valuable research investigating the antiviral effects of berberine hydrochloride (BBH) against Ostreid herpesvirus 1 (OsHV-1) in blood clams (Anadara broughtonii).
The experimental design is well performed, research question is relevant and meaningful.
The article is written in acceptable English without big grammar errors and do not need to be corrected.
The study provides important insights into potential therapeutic strategies for viral diseases in aquaculture.
It is fascinating how berberine hydrochloride (BBH) might help fight viral infections in blood clams. The authors have done solid work investigating a practical problem in aquaculture, and their findings could really help the industry.
While the paper is generally well-written, I have some serious question that should be clarified:
- The finding that 3mg/L works best is really interesting - could we get more discussion about why this might be the optimal concentration, how it was established? Were some pilot experiments conducted? These questions arise because relatively slightly lower (1mg/L), as well as slightly higher doses (6 and 9 mg/L) show more or less similar, but the same lower effects, which is quite confusing and contradictory from a pharmacological point of view, because the therapeutic range of BBH is actually equal to 0 - there is only one point of 3mg/L, 1<3>6. And with a single dose, it is not possible to objectively determine the effectiveness of the therapeutic effect of the substance, in this case the true antiviral effect, since doses less than 3, as well as more than 3, have the same reduced effect.
- The next following questions rising from previous one is about stability and toxicity of BBH.
It would be suggested to discuss:
- potential limitations of the bath treatment approach in practical aquaculture settings;
- a more detailed comparison with other antiviral treatments in aquaculture;
- the economic feasibility or cost-effectiveness of BBH treatment.
Conclusions:
Appropriate revision related to the above points is required.
Reviewer 3 Report
Comments and Suggestions for Authors
General comments:
This study investigates the antiviral potential of berberine hydrochloride (BBH) against OsHV-1 in blood clams, addressing a critical issue in aquaculture. While the findings suggest that BBH can suppress viral replication and improve survival rates, the manuscript requires substantial revisions. The lack of detailed toxicity assessments, such as LD50 or dose-response data, raises concerns about the safety and efficacy of BBH. Additionally, the mechanistic insights into BBH’s effects on viral ORFs remain insufficient, and the connection between viral load reduction and survival outcomes needs clearer explanation. Inconsistencies in nomenclatures and incomplete figure annotations further hinder the overall clarity. These issues must be addressed to strengthen the manuscript’s scientific validity and readability. I recommend a major revision.
Comment 1. (Line 25) The nomenclature in vivo should be italicized to follow standard scientific conventions.
Comment 2. (Line 28-29) The sentence, "The results obtained demonstrated that a 3 mg/L BBH bath immersion significantly suppressed OsHV-1 replication in blood clams," could be revised for clarity and conciseness. The phrase "The results obtained demonstrated" may be redundant. The reviewer suggests rephrasing to:
"The results demonstrated that a 3 mg/L BBH bath immersion significantly suppressed OsHV-1 replication in blood clams."
Comment 3. (Line 92) It is recommended to include a space between numeric values and units (e.g., "0.1 g" instead of "0.1g"). Please ensure consistency and revise this formatting throughout the manuscript, including in figures.
Comment 4. (Figure 1) Based on the results in Figure 1, the authors should clarify the toxicity of BBH in blood clams, such as determining the LD50. This clarification is needed because the 3 mg/L group showed the most protective effect, whereas the 1 mg/L and 9 mg/L groups had less protective results, suggesting potential toxicity. Additionally, in Figure 2, the authors should explain whether the copy number range (1 × 10⁴–1 × 10⁵ copies/ng) corresponds to the LD50, as the 3 mg/L group showed approximately a 46.67% survival rate in Figure 1.
Comment 5. (Figure 4) The images in the two upper panels of Figure 4 are cropped and lack x-axis information. Please provide complete axis labeling for clarity.
Comment 6. (Nomenclature Consistency) The authors should use consistent terminology throughout the manuscript. For example, in some parts, "hours post-infection" is used (e.g., in the legend of Figure 2 and in Figure 5), while in other parts, "hours past-infection" is used (e.g., in Figures 1 and 2). Please ensure uniformity.
Comment 7. (Figures 3–5) The role of ORF41 in viral attachment is noted but not examined in Figures 3–5. Please explain why ORF41 was not included in these analyses.
Comments on the Quality of English LanguageIt is recommended that the manuscript undergo English language editing for improved clarity and readability.
Reviewer 4 Report
Comments and Suggestions for Authors
Dear Editor,
The manuscript entitled “Berberine, a natural compound with antiviral effects against Ostreid herpesvirus 1 infection in Anadara broughtonii” by Hui-Gang Kang et al. presents a study focused on the evaluation of the berberine hydrochloride (BBH) potential as a therapeutic agent against OsHV-1 infection in blood clams (Anadara broughtonii).
Τhe manuscripts’ objectives are quite interesting, the manuscript is well-written and could be accepted for publication after major revisions. My detailed comments for the authors to consider are provided below:
1. Page 3, lines 102-105: please add a reference on starvation effect on drug absorption or provide data to support your claim.
2. Page 3, lines 102-105: Please explain the berberin bath prior infection. If you want to use the berberin as a protective agent instead of theurapeutic agent, you should explain your rational. I cannot understand its use as therapy in the present study and all results should be discussed under that prism (i.e. berberine treatment prior infection-protective effect). Please clarify your rational behind the experimental infection – therapy protocol).
3. Page 4, line 193: Maybe you mean table 1 instead of figure 1?
4. Table 1: In my opinion the table would be better sorted based on the molecular function, to have the gene/ family categories grouped.
5. I would expect to see the BBH in vivo toxicity results before the 3.1. (survival) section. Please add the results or the appropriate reference.
6. Page 7, section 3.3.: Please correct “number relative expression” to “ORF relative expression”.
7. Page 7, section 3.3.: the ORFs expression were only analyzed to 3 time points (i.e. 24, 48 and 72 hpi). Why didn’t you analyze the other time points? Especially the 12 hpi time point should have been analyzed since viral infections appear to induce very early responses in invertebrates. In your discussion you mention 4 hpi as an early time point (page 11, line 349) for infection of the same virus. I stringly suggest to add the 12 hpi in your analysis.
8. Page 7, section 3.3. and figure 2: based on the figure legend, the red color indicates higher expression of the relative expression of 26 OsHV-1 ORFs in the 3 mg/L BBH bath group and control group. However, in the text you mention that at 24 hours post-infection (early stage), the expression levels of OsHV-1 ORFs were significantly reduced in the BBH bath group compared to the control group, therefore I would expect to see it blue. It is a bit confusing to me, could you please clarify?
9. Page 8, lines 276-278 and figure 5: please add the relative expression levels of OsHV-1 ORFs in the experimental group (drug exposure) of blood clams at 48 and 72 hours post-infection too. It would be interesting to see how the variation levels change (or not).
Round 2
Reviewer 1 Report
Comments and Suggestions for Authors
The authors did not provide additional experimental data to support this study.
Reviewer 2 Report
Comments and Suggestions for Authors
Thanks to the authors for their answers and explanations. The manuscript can be accepted for publication.
Reviewer 4 Report
Comments and Suggestions for Authors
I would like to thank the authors for responding adequately in my comments.